# Pathogenesis of Autoimmune Male Infertility: Juxtacrine, Paracrine, and Endocrine Dysregulation

Valeriy A. Chereshnev [1], Svetlana V. Pichugova [1,2], Yakov B. Beikin [2], Margarita V. Chereshneva [1], Angelina I. Iukhta [3,*], Yuri I. Stroev [3] and Leonid P. Churilov [3,4]

1    Institute of Immunology and Physiology, Ural Branch of the Russian Academy of Sciences, 620049 Yekaterinburg, Russia; iip@iip.uran.ru (V.A.C.); ekb-lem@mail.ru (S.V.P.); mcheresh­neva@mail.ru (M.V.C.)

2    State Autonomous Healthcare Institution of the Sverdlovsk Region "Clinical and Diagnostic Center" (GAUZ SO "CDC" Clinical Diagnostic Center), 620144 Yekaterinburg, Russia; inbox@kdc-lab.ru

3    Laboratory of the Mosaics of Autoimmunity, Saint Petersburg State University, 199304 Saint Petersburg, Russia; svetlanastroeva@mail.ru (Y.I.S.); l.churilov@spbu.ru (L.P.C.)

4    Saint Petersburg Research Institute of Phthisiopulmonology, 191036 Saint Petersburg, Russia

\*    Correspondence: ang22748@gmail.com

**Abstract:** According to global data, there is a male reproductive potential decrease. Pathogenesis of male infertility is often associated with autoimmunity towards sperm antigens essential for fertilization. Antisperm autoantibodies (ASAs) have immobilizing and cytotoxic properties, impairing spermatogenesis, causing sperm agglutination, altering spermatozoa motility and acrosomal reaction, and thus preventing ovum fertilization. Infertility diagnosis requires a mandatory check for the ASAs. The concept of the blood–testis barrier is currently re-formulated, with an emphasis on informational paracrine and juxtacrine effects, rather than simple anatomical separation. The etiology of male infertility includes both autoimmune and non-autoimmune diseases but equally develops through autoimmune links of pathogenesis. Varicocele commonly leads to infertility due to testicular ischemic damage, venous stasis, local hyperthermia, and hypoandrogenism. However, varicocelectomy can alter the blood–testis barrier, facilitating ASAs production as well. There are contradictory data on the role of ASAs in the pathogenesis of varicocele-related infertility. Infection and inflammation both promote ASAs production due to "danger concept" mechanisms and because of antigen mimicry. Systemic pro-autoimmune influences like hyperprolactinemia, hypoandrogenism, and hypothyroidism also facilitate ASAs production. The diagnostic value of various ASAs has not yet been clearly attributed, and their cut-levels have not been determined in sera nor in ejaculate. The assessment of the autoimmunity role in the pathogenesis of male infertility is ambiguous, so the purpose of this review is to show the effects of ASAs on the pathogenesis of male infertility.

**Keywords:** male infertility; varicocele; varicocelectomy; spermatozoa; sperm antigens; antisperm autoantibodies; ejaculate; orchitis; autoimmune thyroiditis

## 1. Introduction

In recent years, infertility has become a global health problem [1]. An increasing number of men suffer from impaired fertility, while the incidence of the male factor of infertility reaches 30–50% in infertile couples [2–7]. The reproductive potential of the world male population is declining steadily and there is no tendency for improvement [7–11]. The etiology of male infertility is still a matter of debate among various specialists, but the influence of both exogenous and endogenous factors is noticed, combining negative effects on spermatogenesis and various stages of the fertilization [12,13]. Hence, male infertility is a multifactorial syndrome that includes a wide range of disorders, affecting not only the reproductive system, but also the immunoneuroendocrine apparatus [10,14,15]. The most common causes of male infertility are genitourinary malformations, genetic disorders

(particularly, cystic fibrosis and chromosome aberrations), congenital diseases of the male reproductive system (cryptorchidism, monorchism, phimosis, and hypospadias), varicocele, neuroendocrine disorders and chronic stress, traumas and inflammatory diseases of the reproductive tract both with infectious and non-infectious etiology, lifestyle factors (alcohol, tobacco smoking, drug addiction), and testicular tumors [5,11,14–25]. Immunopathological factors are leading in the pathogenesis of male infertility, essential for its cases regardless of different etiology.

## 2. Sperm Antigens and Antisperm Antibodies (ASAs): Past and Present

In 1901–1903, Russian pathophysiologist Efim S. London obtained the "spermolysins", cytotoxic antibodies against spermatozoa, and predicted their pathogenic role in male infertility and potential use as a contraceptive in andrology [26,27]. At the same time, another Russian scholar Sergei I. Metalnikov obtained the very first experimental model of male infertility using "spermolysins" in animals [28]. Nowadays, 120 years later, ASAs production is considered one of the main mechanisms of male infertility [14,29–32]. The presence of ASAs in the blood and semen of infertile men was first reported by Philip Rűmke in the Netherlands and Leo Wilson in the USA back in 1954, and since then, researchers have focused on them [7,33,34]. However, active clinical pathophysiological studies of autoimmunity as a mechanistic factor contributing to the formation of male infertility began after 1965 [35,36]. In recent decades, an immunological form of infertility has attracted more attention due to the establishment of the role of ASAs as a direct cause of infertility, since higher levels and incidence of ASAs were diagnosed in infertile men compared to healthy ones [13,37–41]. Nowadays, there is an opinion that a pathologically enhanced autoreactive immune response to the antigens of the reproductive system can lead to infertility, and ASAs are valid immunological markers valid for the assessment of impaired male fertility [29,39–41]. ASAs is determined in 10% of infertile men, and the frequency of the immunological form of infertility is 4.5–15% in various populations [2,7,13,41–45].

It is known that autoimmunity to sperm can result from inflammation of testicles (orchitis), both of infectious and non-infectious origin [13–15,20,21,23,32,44,46–52]. Commonly, non-infectious orchitis occurs in traumas (including biopsy, invasive procedures, and surgical intervention, such as vasectomy in 20–30% cases). In general, the development of immunological infertility is associated with any damage to the testicle. According to the "danger hypothesis" by P. Matzinger, such a situation of non-specific inflammation via pro-inflammatory autacoids increases the expression of co-stimulatory molecules on immune cells, thus prolonging the existence of immunosynapses between antigen-presenting cells and autoreactive lymphoid clones. As a result, autoimmunity increases and may reach pathological intensity [53]. Additionally, local damage for a long time was interpreted as a factor destroying the 'immune privilege' of the testis, simplistically attributed to the existence of the anatomical blood–testis barrier. In fact, this point of view is becoming obsolete because the "barrier" is currently regarded not as an anatomical obstacle for antibodies, but as an informational barrier established by local paracrine/juxtacrine action of anti-inflammatory cytokines produced in the testis and downregulation of autoimmunity by androgens [54]. Anyway, local inflammation breaks relative immunological tolerance, which may have serious consequences for the reproductive system [2,7,10,11,22,25,41,46,55–60].

## 3. Immune Privilege of Testes: Informational, Not Only Anatomical Barrier

Testicular immunological tolerance is formed in the perinatal period due to blood–testis and blood–epididymis barriers and immunosuppressive activity of the paracrine and endocrine products of testicle cells [24,54,61–63]. The building of the blood–testis barrier begins with the formation of tight junctions between Sertoli cells, while the first germ cells enter the meiotic phase [2,15,64]. The brief scheme of the blood–testis barrier by Cheng C.Y. and Mruk D.D. [65] is shown in Figure 1.

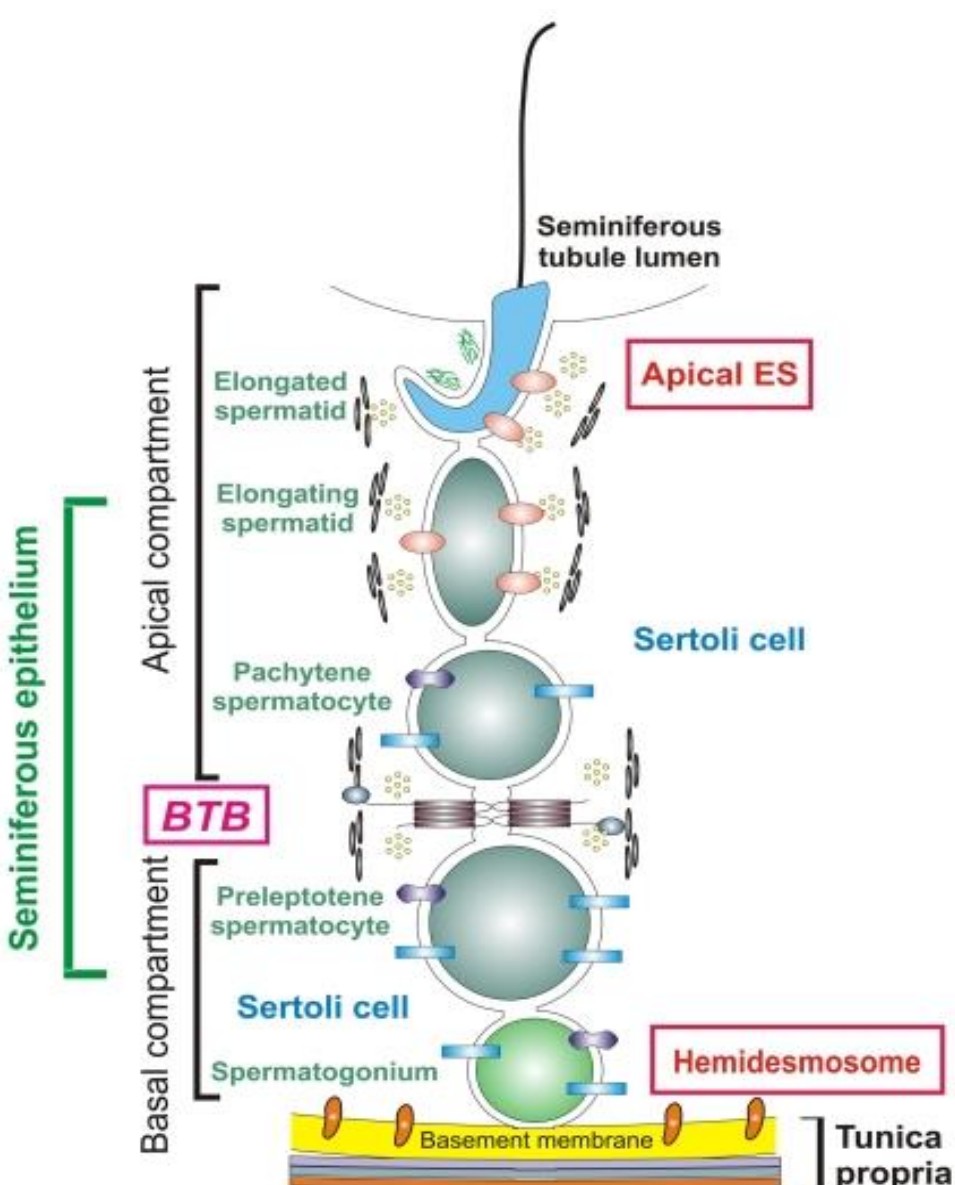

**Figure 1.** The structure of the blood–testicular barrier (BTB) (fragment from Cheng C.Y., Mruk D.D., 2012). The BTB is formed by tight junctions, basal ectoplasmic specialization, desmosome and gap junctions, and the ultrastructural features of the BTB as typified by the actin filament bundles sandwiched between the cisternae of the endoplasmic reticulum and the plasma membranes of two opposing Sertoli cells [65].

Sertoli cells form the basis of the blood–testis barrier; they are located along the length of the seminiferous tubule and relatively isolate the site of spermatogenesis [24,30]. The vascular component of the blood–testis barrier consists of capillary endothelial cells with low permeability, which hampers the passage of lymphocytes and high molecular weight proteins [30]. During puberty, when the germ cells complete their first spermatogenic cycle, differentiating into mature spermatozoa, the testicular immune privilege is put under challenge [24,59]. A lot of new surface molecules are expressed, which, together with the autoantigens of developing spermatozoa, are presented to the immune system. Thus, lymphoid clones may not tolerate them centrally, and their tolerance in these cases critically depends on peripheral local mechanisms [15,66,67]. In addition, in *tubuli recti* and *rete testis*, this shield is absent or fails; hence, the purely anatomical interpretation of the blood–testis barrier is insufficient and outdated [15,24,66]. Young spermatocytes and spermatogonia are located outside the blood–testis barrier and their antigens can be accessed by all elements of the immune system [24]. During spermatogenesis, specific

antigens of meiosis are expressed on cells, and up to 20 additional antigens appear on the membranes of spermatozoa during their passage and maturation in the epididymis [24,30].

The most essential content of the term "blood–testis barrier" is rather informational but does not refer to anatomical protection. Sperm autoantigens are available and recognized by the immune system, but they usually do not activate the pathogenic immune response due to the presence of a peritubular immunoregulatory system, which is the key "non-anatomic" or informational part of the blood–testis barrier [14,30,66].

Several bioregulation mechanisms are involved in establishing peripheral tolerance in the testicles: anti-inflammatory polarization and decreased reactivity of resident macrophages, inhibited proinflammatory and enhanced tolerogenic cytokines induction by testicular androgens (especially during and after puberty), anti-inflammatory cytokines production by local T-regulatory cells and somatic cells, antigen-specific immune suppression by tolerogenic local dendritic cells, and dormant local mastocytes [66,67].

The blood–testis barrier carries out immunoregulatory juxtacrine and paracrine control by stimulating the release of specific immunoprotective substances from Sertoli cells and Leydig cells, which suppress blast transformation of lymphocytes and prevent lysis of spermatozoa [30]. Some studies have shown the immunosuppressive influence of Sertoli cells in vitro on activated T lymphocytes [24]. The phagocytic activity of Sertoli cells consists in the degradation of spermatozoa and their antigenic components, normally without antigen presentation. Sertoli cells interact with syngenic T-lymphocytes in order to establish testicular immune privilege, and this interaction is HLA restricted. They impede penetration and survival of T effectors and Th17, while at the same time facilitating T regulators [30,66,68]. Normally, T regulators prevail among local lymphocytes over T effectors and block excessive autoimmunity in spite of existing autorecognition [65,66]. However, testicles are normally available for lymphocytes of all subpopulations. Moreover, CD8+ T effectors are essential for the physiologic regulation of testicular germ cell population balance via their apoptogenic signals; hence, if the old concept of absolute immune isolation of gonads is true, normal development and renewal of germ cells would not be possible [69].

Thus, the immune mechanisms of the testicles are physiologically and immunologically prepared both by local (para- and juxtacrine) and systemic androgen-mediated endocrine regulation to protect sperm autoantigens from the destructive autoreactive response, and the presence of a blood–testis barrier makes the testicles an immunologically privileged site [59,70].

## 4. Role of Epididymis

ASAs production may depend on events happening in the region of the epididymis, which is protected by the hemato-epididymal barrier, first described by D.S. Friend and N.B. Gilula in 1972 [70,71]. The epididymis is a highly specialized organ involved in the maturation, transport, protection, and storage of sperm before ejaculation. The blood–epididymal barrier anatomically consists of tight junctions between epididymal cells, and transporters located along their surface, which regulate the bidirectional movement of molecules, promoting sperm maturation and establishing the relative separation of sperm antigens from the cells of the immune system. Although the hemato–epididymal barrier is architecturally more complicated than the blood–testis one, it is anatomically even less effective, which makes the epididymis generally more susceptible to immune influences in comparison with testicles [70]. Maturation of spermatozoa occurs when they pass through the epididymis, accompanied by surface fixation of many proteins synthesized by epididymal cells. Probably, all the above mentioned makes it possible to mark the epididymis as a key site of ASAs generation elicitation [46,70].

The formation of ASAs in men may be associated with impaired immunoregulatory mechanisms or the development of pathological processes, increasing the degree and duration or auto-presentation in the testicles and/or epididymis. A lack of local and

systemic tolerogenic influences and/or enhancement of local/systemic adjuvant effects may shift the balance towards abnormal excessive anti-sperm autoimmunity.

## 5. Targets of ASAs: Assorted Mosaic

The immunological form of infertility is diagnosed if ASAs with immobilizing or agglutinating properties are found in the patient's blood or fluids of the reproductive tract [29,72]. The target sperm autoantigens are specific proteins related to fertilization and fertility, and each of them has a unique structure and is synthesized by different cells of the reproductive system [7,32,72–74]. Currently, the most studied among them are the proteins YWK-II (protein of the equatorial sector of the sperm head), BE-20 (protein of the epididymis), rSMP-B (sperm tail antigen), BS-17 (calpastatin), ACTL7a (an actin-like protein), BS-63 (nulloprotein of testis), HED-2 (a component of Sertoli cells), EP-20 (epididymal glycoprotein of 20 kDa molecular mass), NASP (autoantigenic nuclear protein), FA-1 (specific fertilization antigen), YLP12 (dodecamer peptide specific for acrosomal region of human sperm cells), HSP 70 and HSP 90 (heat shock proteins), and many others [24,29,72–75].

Interestingly, it appeared that in many cases, ASAs are addressed not to genuine autoantigens of spermatozoa, but to immunodominant autoantigens of prostasomes (extracellular vesicles of 40–500 nm in diameter, which are normally secreted by the prostate gland epithelial cells into seminal fluid and used to fix on spermatozoa). Among them, the most frequent targets of ASAs are clusterin and prolactin-inducible protein (PIP), with 10 other prostasome proteins revealed sporadically [76]. The presence of anti-PIP among ASAs is, in our opinion, especially remarkable (see Section 9 data on hyperprolactinemia as a factor increasing the risk of autoimmunity).

Sperm antigens are localized on the head of spermatozoa, in the acrosome, as well as on the flagellum and regulate sperm motility, providing capacitation and initiation of acrosomal response [72,77].

ASAs (Ig A, M, and G) can be found in blood serum and ejaculate (only IgA and IgG, but not IgM due to its size) [2,11,39,41,42,46,74–78]. In addition, ASAs can be addressed to surface or intracellular sperm antigens [24]. IgA and IgG can passively diffuse into the reproductive tract, but IgA can also be actively secreted by germ cells. Epithelial cells produce a secretory component that acts as a regulatory transport protein for IgA. Antigenic epitopes in infertile men bind more avidly to local IgA than to IgG, and IgG is less reactive against sperm antigens [30]. The detectable amounts of ASAs IgA may be absent in blood serum; however, their local presence in the genital tract can lead to dysfunction of spermatozoa [24]. It is known that excessive binding of ASAs by spermatozoa may diminish their levels detected in seminal plasma [72]. The degree of fertility impairment depends on the class of autoantibodies, their amount, specificity, and density of their coverage on the sperm cell surface [2,7].

Regardless of the type of ASAs, they can be fixed on different parts of the sperm cells depending on their specificity. Usually, autoantibodies that react with sperm surface antigens are agglutinins or immobilisins and cause different types of spermatozoa agglutination (head + head, head + flagellum, flagellum + flagellum) [30]. Presumably, those ASAs that interact with the antigens of the membranes of vitally important antigens of the sperm cell will be of clinical value [24]. Thus, the attachment of autoantibodies to the flagellum of the spermatozoa will lead to impairment of cell mobility, while fixation on the head will lead to impaired penetration of the spermatozoa into the cervical mucus [24,30]. Autoantibodies bound to the acrosomal region can interfere with the acrosomal reaction, leading to the occlusion of receptors necessary for attachment to the transparent membrane of the ovum, thus preventing fertilization [30,32]. ASAs fixation on sperm can not only lead to agglutination and immobilization [14,29], but also has a cytotoxic effect mediated via complement and/or macrophages or other K cells [2,41,79]. Binding of IgG and complement proteins initiates C-3-mediated interaction of spermatozoa with polymorphonuclear cells and inactivation of spermatozoa through the deposition of the membrane attack complex (MC5b-9) of the complement. Cytotoxic antibodies, especially in seminal plasma, can

cause a premature acrosomal reaction, since a large number of antigens are concentrated on the acrosome as a membrane structure [15,30,64]. The combined effects of various ASAs on different components of spermatozoa can also lead to immunological infertility [24].

It has been noticed that the quality of sperm deteriorates significantly in the presence of ASAs [80]. The effect of ASAs on reproductive function can be realized in various ways: impaired spermatogenesis, sperm agglutination, decreased motility, impaired penetration of spermatozoa into cervical mucus, impaired acrosomal reaction, obstruction of ovum fertilization, impaired embryo implantation, and deranged early development of zygote [2,29,32,39,55,70,78,81–85]. Due to the relatively small number of cases involved in studies, no statistically significant correlation of infertility has been established so far with antibody isotypes, titers of ASAs, age of men, or sperm count [24,30].

ASAs-mediated infertility should be suspected if sperm agglutination and motility impairment are diagnosed by semen analysis in the absence of leukocytospermia and infection [24,30,86].

It is also interesting that some articles show that vitamin D binding protein has a molecular similarity to ASAs, and both low (<50 nmol/L) and high (>125 nmol/L) concentration of vitamin D are associated with a decreased number and quality of spermatozoa in semen [87].

In spite of all data cited above, the link between the presence of ASAs in men and infertility continues to be disputed, and it is unclear whether ASAs adversely affect the outcome of in vitro fertilization (IVF) or results of intracytoplasmic sperm injection (ICSI) [88].

Studies on the likelihood of fertilization after IVF have yielded conflicting results, with some data showing negative effects of ASAs, while others have shown no negative effects at all. Perhaps, that is due to differences between ASAs of various molecular specificity, which is not registered in most assays.

Studies on ASAs and post-ICSI pregnancy rates have generally shown that ASAs do not affect post-ICSI pregnancy rates [59,88–91].

It should be noted that ASAs in the immunological form of infertility are formed not only in men, but also in women.

Several researchers showed that among women with unexplained infertility, ASA activity was detected in the cervical mucus in more than 10% of cases. Moreover, ASAs from female serum could inhibit the fertilization of viable human eggs by spermatozoa, and the fertilization rate was only 15% for female patients who had significant blood titers of IgG and IgA ASAs. Later in vitro experimental results confirmed that high-titer ASAs IgG in female serum could effectively inhibit fertilization [92]. R. Bahraminejad et al. [93] showed that prostitutes have significantly higher titers and a wider target spectrum of ASAs than women having one sexual partner, and the intensity of antisperm autoimmunity in this group correlates with the percentage and timing of infertility.

As regards natural conception, ASAs may be a possible cause of decreased reproductive potential in women with secondary infertility. Particularly, increased miscarriage rates in women with ASAs were demonstrated by some authors [94–96]. However, there is evidence that in women with polycystic ovary syndrome or endometriosis, ASAs do not play a significant role in the occurrence of infertility, and also do not cause miscarriage in the first trimester [97–100].

## 6. ASAs and Varicocele

As it was mentioned above, any non-specific damage increases the intensity of auto presentation in damaged tissue, especially if it is accompanied by inflammation [53]. Varicocele is characterized by varicose veins of the spermatic cord; its incidence among males is 15–40%, which makes it the most common andrological entity [81,83,101–103]. Being an aggressive form of orchopathy, varicocele causes highly increased risk of infertility [1,101,103–107].

Varicocele's association with infertility is based on the number of observations by different authors, who diagnose it more often among infertile men than in the general

population. Additionally, it is accompanied by abnormalities in the spermogram parameters [104]. The peak of varicocele falls on puberty. It occurs in 15–19% of adolescents of this age group and can limit the future reproductive potential of this cohort [19,103].

The pathogenesis of varicocele is multifactorial [84,103]. It causes congestion of local blood circulation. Hypoxic damage of the testicular parenchyma, relative scrotal hyperthermia, venous hypertension, and passive hyperemia reflux of metabolites and bioregulators (catecholamines) from the kidneys and adrenal glands, as well as hypoandrogenism are considered as the main mechanisms resulting in damage to the blood–testis barrier [7,15,58,81–83,103]. Disruption of the transport of water, lactate, and other substances in Sertoli cells occurs [64,83,108,109]. All these mechanistic links in the pathogenesis of varicocele lead to an increase of oxidative stress, decrease of tolerogenic influences, and increase in ASAs production [59,109]. Additionally, a decrease in the expression of E-cadherin and alpha-catenin at the junctions of Sertoli cells was found in varicocele, which can increase the permeability of the blood–testis barrier [83,110]. As already noted, ASAs are often found in infertile men with varicocele [7,41,59,81]. Immunological infertility was registered in 15–28% of men with varicocele [78,80,103,111]. In addition, B.R. Gilbert et al., as early as 1989, established that ASAs are present in greater quantities in infertile men with varicocele than in infertile men without it, and this was later confirmed by A.M. Al-Daghistani et al., and a number of other researchers [30,58,83,103,109,111–115].

The presence of ASAs in varicocele is accompanied by a deterioration in sperm parameters. In particular, there is a decrease in progressive sperm motility (asthenozoospermia is diagnosed most often), as well as a decrease in the concentration and total number of spermatozoa, with violation of the acrosome reaction [39,40,58,81,82,103,109–115].

However, on the contrary, Kanevskaya T.A. et al. showed that ASAs did not increase notably in infertile men with varicocele compared with infertile men without it. According to their study, ASAs did not correlate significantly with male infertility in varicocele [116]. Several other groups also demonstrated that varicocele did not affect the concentration of ASAs, and ASAs, in turn, did not affect the parameters of sperm in varicocele. Moreover, concentrations of ASAs in fertile or infertile varicocele patients were close [78,83,111,113,117]. In a Finnish cohort of 508 infertile men, the levels of ASAs among those having fresh or anamnestic varicocele were even lower than in infertile men without varicocele [118]. Hence, there is an assumption that autoimmune mechanisms are not the single or even the main cause of infertility in varicocele. Presumably, ASAs' role is permissive and accompanied by additional damaging factors, for example, trauma, infection, and resulting inflammation [46,78,82,83,111,117].

## 7. Sperm Autoimmunity and Varicocelectomy: Conflicting Data

Varicocele is traditionally considered a potentially curable cause of male infertility, and varicocelectomy still serves as the gold standard of treatment; however, surgical intervention often fails to restore fertility and improve semen analysis, so the outcome of the operation remains poorly predictable [24,51,76]. Earlier, ASAs were not considered as a factor that negatively affects the result of surgical treatment, but nowadays, many researchers have reported an increase of ASA levels after surgical intervention [42,43,50,78,80,86,116,119–121].

The results of varicocelectomy may be equivocal. Several groups found that in those cases when the patients operated upon did not show an improvement in spermogram parameters (still displaying impaired sperm motility), there were higher levels of ASAs registered. In the long-term postoperative period, an improvement in spermogram parameters was noticed mainly in patients initially negative for ASAs [81,82,111–113]. In a number of studies, after the surgery, an increase in the level of ASAs was recorded, but it did not cause any negative effect on sperm parameters [78,84,122]. Nevertheless, one study of 2016 noted after varicocelectomy, a decrease in ASAs compared to their initial levels, and an improvement in spermogram, primarily, in sperm motility was observed [55]. The majority of works did not register any significant differences between the concentration

of ASAs before and after varicocelectomy, including data of the long-term postoperative period [42,83,113,115].

In general, in those patients without ASAs, varicocelectomy most often leads to an improvement in sperm parameters. However, ASA-positive patients who undergo surgical treatment have less successful results. Most often, varicocelectomy increases or does not alter the titer of ASAs, but there are few articles describing its post-operative decrease.

## 8. Anti-Sperm Autoimmunity and Infection: Essential Trigger

As already noted by some researchers, infection of the reproductive tract is one of the common causes of male infertility, potentially curable, but, nevertheless, leading to a decrease in sperm quality due to various mechanisms. One of them is oxidative stress, causing fragmentation of sperm DNA and initiating apoptosis [48,60,123–125].

Additionally, any local infection provoking inflammation shifts the balance of pro- and contra-autoimmune bioregulators towards excessive autoimmunity, increasing the presence and influences of T effectors and diminishing the number and activities of T regulators and/or changing the trend of macrophages' polarization [7,14,53,54,59,66,67]. Many molecules expressed during the infectious process are in fact endogenous adjuvants enhancing auto presentation and autoimmunity. Particularly, men who had leucocytosper-mia associated with bacteriospermia demonstrated an elevated expression of Toll-like receptors 2 and 4 and showed a significant increase in oxidative stress indices, immune response against spermatozoa, and spermal dysfunction [29].

In general, infectious processes can disrupt local immune regulation, and damage spermatozoa directly or through an inflammatory reaction, leading to the formation of ASAs [121]. It was found that in more than 40% of cases, ASAs to sperm surface antigens were found in infected patients, which is a significantly higher rate than among uninfected infertile and fertile men [125].

The human body is a habitat for a huge number of different types of bacteria, viruses, fungi, and parasites, which have undergone a long co-evolution with us. Hence, the phenomenon of molecular mimicry, used by microorganisms to suppress the host's immune response, gives them the ability to initiate autoimmunity by mimicking host proteins, including sperm antigens; therefore, molecular mimicry may be a crucial mechanism causing autoimmune infertility of infectious etiology [126].

The idea of molecular mimicry was coined in biology by a Russian zoologist Konstantin S. Merezhkovsky in the very beginning of the 20th century [127], and in the last decades, it has enrolled a lot of proponents among immunologists [128].

Indeed, ASAs are found in people who do not have any obvious anamnestic reason for their production [7]. Either that fact should be interpreted as a witness for the doctrine of physiological autoimmunity and autoimmune regulation of morpho-functional processes in testes [129], or, alternatively, the reason is a hidden immune response against some microorganisms sharing epitopes with spermatozoa [128]. For example, a specific immune response to *Chlamydia trachomatis*, the causative agent of the most common sexually transmitted infection, can lead to inflammation and impaired fertility by DNA fragmentation and also by activating immune responses to an epitope of a heat shock protein, shared by *Chlamydia* and human sperm cells, which is accompanied by high levels of ASAs [21,29,124,126]. The epitopes shared with sperm cell antigens have been found in many microbial antigens from *Escherichia coli*, *Bacillus sp.*, *Staphylococcus aureus*, *Streptococcus pyogenes*, *Streptococcus agalactiae*, and many viruses [7,29,125,126,130]. *Helicobacter pylori* is also considered as a possible initiator of autoimmunity against spermatozoa, as since spermatozoa are the only human cells possessing flagella, the possibility of their homology with bacterial flagella cannot be ignored [126]. Thus, not only pathogens of sexually transmitted infections, or other pathogenic microorganisms, but also bacteria of opportunistic flora can initiate autoimmune reactions against spermatozoa. However, D. Kanduc emphasized recently [128] that virtually all microbes and viruses have some epitopes shared with some human peptides. It does not necessarily mean that such a

microorganism will inevitably provoke an autoimmune process in all humans. The point is that individuals with different HLA haplotypes may process the same proteins differently, slicing them on various peptides and responding with different strengths in the context of an individual set of major histocompatibility complex antigens. For example, mumps is a well-known cause of viral orchitis. However, nevertheless, the levels of ASAs in infertile men who experienced mumps in anamnesis appeared to be lower than in those infertile men who never suffered from this infection [118].

Anyway, if the etiology of male infertility is anamnestically related to infection, it does not mean that autoimmune mechanisms are not involved in its pathogenesis. Both the "danger hypothesis", postulating the adjuvant effect of infectious inflammation on auto presentation and autoimmunity [53], and the molecular mimicry concept allow anti-self B and T effectors to start their job after help from anti-alien T helpers [126,128], which gives a substantial theoretical basis for such a scenario.

The current pandemic of the new coronavirus infection COVID-19 may also alter male fertility, addressing autoimmune mechanisms. It has been demonstrated that Sertoli cells, Leydig cells, and spermatogonia all express the ACE2 receptor, offering a gate for SARS-CoV-2 penetration [131]. Moreover, cases of orchitis and orchiepididymitis were registered in males after SARS-CoV infection in the past, and after COVID-19 [132–134].

## 9. Systemic and Multiorgan Autoimmune Diseases and ASAs

Systemic and multiorgan autoimmunopathias can adversely affect male fertility. In particular, autoimmune thyroiditis, especially in advanced cases, leading to hypothyroidism, has a profound negative effect on the male reproductive system. The reasons are not only some shared thyroid and sperm antigens, like human meichroacidin [135], but most probably, the systemic action of endocrine disorders in Hashimoto's thyroiditis. Thyroid hormones via genetic, epigenetic, and non-genomic mechanisms, in particular, by paracrine action on Sertoli cells, Leydig cells, or spermatozoa, are involved in maintaining sperm quality, with $\alpha$ and $\beta$ receptors for thyroid hormones expressed in the human testicles [136]. It has been shown that thyroid hormones act directly on calcium channels, providing an increase in calcium influx and cAMP synthesis, activating protein kinase A, which causes movements of spermatozoa, and leading to their hyperactivation [137]. Moreover, thyroliberin, produced in Hashimoto's thyroiditis patients as a compensatory response to coming hypothyroidism, also has considerable prolactogenic activity. Because of this, advanced cases of Hashimoto's thyroiditis almost always cause hyperprolactinemia [138]. Hyperprolactinemia suppresses androgen production (which decreases androgen influences essential for the establishment of testis immune privilege) [139]. Further, prolactin acts as a potent autoimmunity stimulant both on endocrine and paracrine levels [140]. As it was mentioned in Section 5 above, PIP of prostasomes attached to spermatozoa is one of the proven targets of ASAs [76]. Thus, the vicious circle in Hashimoto's thyroiditis is formed: the more destructive the autoimmune process, the deeper hypothyroidism, the greater degree of hyperprolactinemia, and thus stronger autoimmunity development [138].

This complex of endocrine disorders makes Hashimoto's thyroiditis an important risk factor for male infertility. In our studies, it was found that the level of anti-thyroid peroxidase autoantibodies (markers of this disease) correlated positively with pathozoospermia [141]. Long-term non-compensated hypothyroidism leads to pituitary dysfunction, hyperprolactinemia, and hypogonadism [142]. Ultimately, hypothyroidism in autoimmune thyroiditis negatively affects spermatozoa morphology and sperm motility and deteriorates the parameters of spermiogram. Levothyroxine therapy reverses these abnormalities [129,143,144]. Thyroid hormones in the treatment of Hashimoto's thyroiditis act not only as a replacement therapy, but also as immunomodulating agents, because they suppress hyperprolactinemia with its pro-autoimmune effects and even facilitate the apoptosis of lymphoid clones [145,146]. Autoimmune thyroiditis often combines with other autoimmune diseases, sometimes, with autoimmune orchitis. For example, amiodarone (anti-arrhythmic medicine with a huge content of iodine) induces both thyroid and testicu-

lar autoimmunity [147]. In Hashimoto's thyroiditis, an increased level of autoantibodies to steroid-producing cells may be observed, with decreased serum testosterone, which causes altered spermatogenesis, resulting in a low concentration of spermatozoa, a decreased number of progressive motile spermatozoa, and morphologically abnormal spermatozoa in semen [148]. If autoimmune orchitis was caused by systemic vasculitis, like in systemic lupus erythematosus (SLE), the patients may also be ASAs positive. ASAs have been reported in almost half of male SLE patients. Testicular inflammation may induce a T cell response with pro-inflammatory cytokine production and resulting blood–testis barrier alteration, ASAs production, and apoptosis of spermatozoa [14].

Bulgarian researchers reported multiple targeted autoimmunity including ASAs and various rheumatological marker autoantibodies in 55% of infertile couples for both partners [149]. There are also studies showing an increased incidence of ASAs and secondary autoimmune orchitis in males with various systemic autoimmune diseases, such as polyarthritis nodosa, Behçet's disease, rheumatoid arthritis, and Henoch–Schönlein purpura. The overall frequencies of acute orchitis and ASAs in rheumatic diseases are 2–31% and 0–50%, respectively, which is significantly higher than in healthy controls [150,151].

## 10. Methods of ASAs Detection

ASAs in men can be determined in blood serum, seminal plasma, or on the surface of spermatozoa using various tests. An ideal test for the diagnosis of autoimmunity against spermatozoa should detect the presence of ASAs, their localization, and isotype, with high sensitivity and specificity. The existing tests are compared and described in detail elsewhere [24]; a brief synopsis follows.

There are several tests for detecting ASAs, described in detail by Hulusi B. et al. [24].

The presence of ASAs in blood serum or semen can be revealed by an agglutination test using washed motile sperm from a healthy donor, added to diluted patient samples. In addition, known positive and negative controls are required. The test results are evaluated using light microscopy and the test is considered positive in cases where there is a clear agglutination of sperm/A titer of 1/32, which is considered of clinical value. This test allows determination of the type of agglutination but does not provide quantitative data and requires a healthy sperm donor.

Another available test for detecting ASAs is the sperm immobilization assay, a complement-dependent analysis of sperm motility. ASAs interact with sperm antigens activating the complement system, which leads to a violation of the permeability and integrity of the sperm membrane, followed by a loss of mobility and cell death. The results are also recorded using light microscopy. The disadvantages of the method include the lack of quantitative data. Besides, only immobilizing antibodies are determined. Moreover, IgA is not detected at all since they do not fix complement.

The MAR (mixed agglutination response test) is used to detect ASAs with erythrocytes, on which IgG to sperm antigens are conjugated. The method is semi-quantitative, evaluated by light microscopy. The test is easy to perform, but it has several weak points: the method cannot be used in patients with oligozoospermia and asthenozoospermia; motile spermatozoa are required, hence immobilizing antibodies are not evaluated; an excess of erythrocyte prevents correct assessment of the sites of binding with spermatozoa; and the presence of mucus, microorganisms, and non-immunoglobulin proteins interferes and can cause false positive results [24].

A modification of the MAR test is the sperm MAR test, in which latex beads conjugated to IgG or IgA are used instead of erythrocytes. When a highly specific IgG is fixed on the beads when spermatozoa are added, the beads are attached strictly to the area where the antigen is localized. The sperm MAR test for IgA detects only secretory IgA in semen. Nevertheless, among all the ASAs tests, the MAR test showed the highest performance. It is considered positive when more than 40% of spermatozoa bind to latex particles when tested for IgG [24].

For IBT (indirect beads tests), polyacrylamide beads coated with a specific anti-immunoglobulin are used. The direct IBT test is used to determine the ASAs in semen, and the indirect test can be used in seminal plasma with a small number of motile sperm. The binding of the beads to sperm is assessed using light microscopy. The principles of IBT are shown in Figure 2 [152]. The test is considered significant if 50% or more of motile spermatozoa are covered with beads. Compared to the tests described above, the IBT test has a number of advantages, such as the ability to determine the isotype of ASAs and their localization, and it also has high sensitivity and specificity [24,152,153].

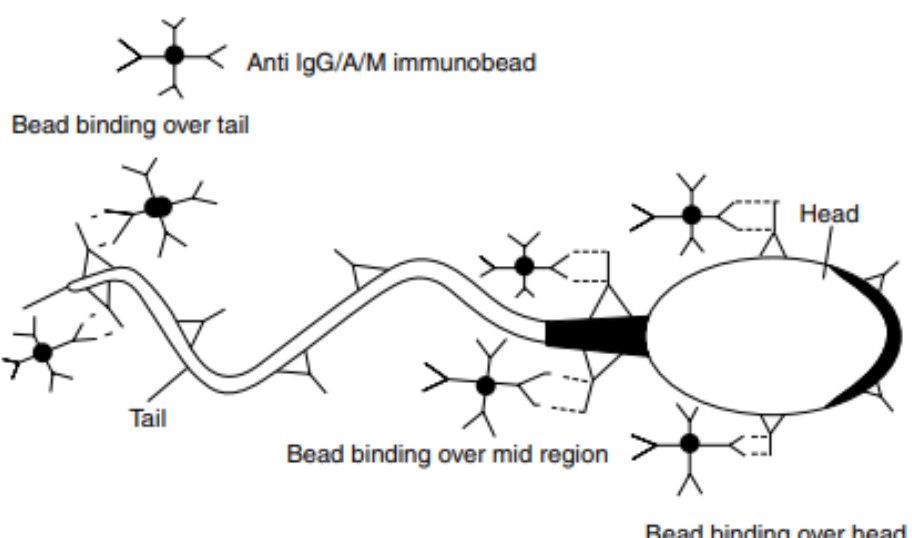

**Figure 2.** The scheme of the immunobead test (IBT) (fragment from Sikka, S.C., Hellstrom, W.J.G., 2019). The immunobeads are microscopic polyacrylamide spheres that carry covalently bound rabbit antibodies directed against human immunoglobulins. Sperm and beads are mixed, and the suspension is observed by microscopy for agglutination of sperm and beads. By using beads coated with Ig-class-specific antibodies, one can identify the different antibody classes involved (IgG, IgA, IgM) [152].

The WHO recommended both MAR and IBT tests as screening methods for ASAs determination [7,11,35,119–121,154]. The MAR test is considered sufficient because IgA almost never occurs without IgG, and, as it was already mentioned, there is a variant of the Sperm MAR test that can detect IgA associated with sperm [24].

In addition, a common and effective test for the diagnosis of immunological infertility is the enzyme-linked immunosorbent assay (ELISA), which allows the determination of IgA and IgG in blood serum and on the surface of spermatozoa, and many researchers prefer to use this method for diagnosis, despite the WHO recommendations [10,44]. The essence of the method lies in the fact that ASAs to specific immunoglobulins are covalently bound to enzymes and antibody-enzyme-Ig complexes are detected by adding an enzyme substrate, which leads to a color change. Thus, the ASAs level can be quantified. A significant disadvantage of the method is the use of fixed sperm, since fixation can lead to a violation of the integrity of the sperm membrane and intracellular antigens will participate in the reaction, altering the results. Additionally, this technique does not allow check the isotype of the antibodies or their localization to be checked and has relatively lower sensitivity and specificity [24].

Another common method of ASAs identification is flow cytometry. The antibody-coated spermatozoa are incubated with fluorescent antiglobulin, followed by detection with a magnet or laser in a continuous flow. The undoubted advantages of the method include the ability to determine the isotype of immunoglobulins, and quantitative assessment of spermatozoa positive for antibodies. However, there is a high probability of detachment of

the antigen-antibody complex from the sperm surface during diagnostic manipulations, leading to a distortion of the result [24].

Few other methods of ASAs evaluation are in use, almost exclusively in research practice, including radioimmunoassay (radioactively labeled antiglobulin assay), immunofluorescence method, affinity chromatography, and immunoblotting, but they are not used in routine [24,59].

Recently, computer tests were suggested for detecting ASAs without any subjective judgment, and the results were the same as in standard tests. This modification of ASAs analysis has a great future perspective [152].

Because of the great variety of the methods used in ASAs studies, there is considerable discrepancy in the results. It makes comparative studies difficult and the prevalence of ASAs in the population is still not understood accurately [15,24,41,153–157].

## 11. Conclusions

There are several reasons for the heterogeneity of the data obtained on ASAs by various laboratories. First, the sample sizes are noteworthy. In most studies, they rarely exceed 100 cases. Only in a few cohorts were more than 1000 men from infertile couples examined, but there was no comparison group that was adequate in terms of the number of patients (with varicocele or with autoimmune diseases). Secondly, the groups are heterogeneous in terms of age, race, degree of varicocele, duration of surgical correction, anamnestic data, and comorbidity. Thirdly, a number of authors determined ASAs only in the blood serum or only in the ejaculate, and only few have determined ASAs in both biological fluids. Researchers have identified different Ig isotypes, either in isolation or in various combinations.

Until now, there has been no standardized and globally accepted test for the determination of ASAs, and the question remains unresolved regarding which method is optimal for screening and for final characterization of ASAs. A large number of sperm autoantigens have been described. The ASAs may have agglutinating, immobilizing, and cytotoxic activities depending on their precise specificity. Autoimmune infertility most likely occurs due to the combined effect of various autoantibodies to many sperm antigens. It is definitely one of the reasons for the controversy over the relationship between ASAs and immunological infertility in men.

It has not been agreed which cut levels are thresholds of clinical value of various ASAs, neither in serum nor in ejaculate, which leads to a lack of consensus on the clinical significance of positive ASAs tests.

At the moment, male infertility remains one of the leading reasons for the decline of demographic indicators worldwide. The presence of ASAs in high titers is associated with infertility in men. However, nowadays, there is no standardized method for detecting ASAs in various parts of the reproductive system.

**Author Contributions:** Conceptualization, V.A.C. and L.P.C.; writing—original draft preparation, S.V.P., Y.B.B., M.V.C. and A.I.I.; writing—review and editing, L.P.C. and Y.I.S.; supervision, V.A.C. and L.P.C.; project administration, V.A.C. and L.P.C.; funding acquisition, V.A.C. and L.P.C. All authors have read and agreed to the published version of the manuscript.

**Funding:** This research was carried out within the framework of the state assignment of the Institute of Philosophy of the Ural Branch of the Russian Academy of Sciences (topic No. AAAA-A18-118020590108-7). This research was supported by a Grant from the Government of the Russian Federation for support of the studies conducted under the supervision of leading scientists (Contract No. 14. W03. 31. 0009 of 13 February 2017).

**Institutional Review Board Statement:** Ethical review and approval were waived for this study, because it is review of literature.

**Informed Consent Statement:** Not applicable.

**Acknowledgments:** Authors are grateful to Andrew I. Gorelov for valuable consultations as regards to the clinical aspects of the article concept.

**Conflicts of Interest:** The authors declare no conflict of interest. The funders had no role in the design of the study; in the collection, analyses, or interpretation of data; in the writing of the manuscript, or in the decision to publish the results.

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
