# Peer review of "Pathogenesis of Autoimmune Male Infertility: Juxtacrine, Paracrine, and Endocrine Dysregulation"

_pathophysiology, doi:10.3390/pathophysiology28040030_

Round 1

Reviewer 1 Report

This is a well written review and with the following minor revisions, this manuscript could be more improved.  Some comments on mechanistic parts are listed below;

  1. On the production of ASAs, it might be also important to know how immune system recognizes the antigen and how the system responses to the antigen. In this aspect, immune regulatory molecules and regulatory immune system (including DC and T cell) need to be addressed.

  1. For the ASAs positive patients (or some animal models), some therapeutical approaches modulating immune system could be scientifically important, as via evaluating the responses to the therapeutic approaches, we might get insight into more mechanistic understanding.  If authors agree with this, short introduction on this part could be helpful for readers.  There are some papers stating that Vitamin D3 could be effective to reduce ASAs, which is not clearly widely understood.

Author Response

Hello, thank you so much for your comments, they are very useful. I have tried to expand the review a little, basing on your notes. I have added links #75 (this article contains the table with some sperm antigenes)  and #85 (about effects of vitamin D3 on sperm quality).

Reviewer 2 Report

This is an interesting review on the pathogenesis of autoimmune male infertility. The manuscript is well written. I only have a few minor comments.

Title

The article title should clearly indicate that it is a narrative review.

Introduction

The aim of the review should be added

Pag. 2, Line 62

Reference # 35 should be checked

Pag. 3, Figure 1

Was the copyright clearance obtained?

Pag. 6, Line 238

A recent review could be cited (Silva et al, 2021; doi: 10.1530/REP-21-0123).

Pag. 6, Line 259

The authors should consider that increased miscarriage rates in women with ASA were demonstrated by some authors (e.g. Witkin et al, 1988, DOI: 10.1016/0002-9378(88)90776-4; Naz, 1992, doi: 10.1095/biolreprod46.1.130).

Pag. 9, Line 393

After '... CoV-2 penetration.' a reference should be added (e.g. Zupin et al. 2020, doi: 10.1007/s10815-020-01917-0)

Pag. 11, Line 497

There is a new edition of WHO laboratory manual for the examination and processing of human semen

https://www.who.int/publications/i/item/9789240030787

Author Response

Hello. Thank you for all of your comments, they are very accurate. I have corrected the title and abstract, added recommended references (in lines 238, 259, 393). Also I have checked the references #35 and #149 (in new variant #153 - WHO manual). All copyrights were reserved.